# Comparing Musculoskeletal Injuries across Dance and Gymnastics in Adolescent Females Presenting to Emergency Departments

**DOI:** 10.3390/ijerph20010471

**Published:** 2022-12-28

**Authors:** Yixuan A. Pei, Mattia A. Mahmoud, Keith Baldwin, Corinna Franklin

**Affiliations:** 1Perelman School of Medicine at the University of Pennsylvania, Philadelphia, PA 19104, USA; 2Children’s Hospital of Philadelphia, Philadelphia, PA 19104, USA; 3Department of Orthopedics & Rehabilitation, Yale School of Medicine, New Haven, CT 06510, USA

**Keywords:** dance, gymnastics, injury patterns, prevention strategies

## Abstract

(1) Background: Studies have yet to identify if there are any differences in musculoskeletal injury patterns between dance and gymnastics. This study aimed to determine if different injury patterns exist in adolescent females participating in those two popular sports. (2) Methods: A cross-sectional study was conducted using data collected from patients presenting to U.S. emergency departments participating in the publicly available, de-identified U.S. Consumer Product Safety Commission’s National Electronic Injury Surveillance System (NEISS) throughout the year 2020. Regression analyses were performed to explore if injury patterns were predictive of gymnastics or dancing participation. (3) Results: 518 adolescent females with dance-related injuries and 597 adolescent females with gymnastics-related injuries in 2020 were examined. Strain/sprains (33.3%) and fractures (37.3%) were the most reported dance- and gymnastics-related diagnoses, respectively. Participants were 74% less likely to have a strain/sprain diagnosis in gymnastics compared to dance (OR = 0.26, 95% CI [0.18, 0.38]) and were 3.84 times more likely to have a fracture diagnosis from gymnastics compared to dance (OR = 3.84, 95% CI [2.67, 5.57]), even after adjusting for body party injured. (4) Conclusions: Dance is associated with more sprains while gymnastics typically resulted in a greater likelihood for fractures.

## 1. Introduction

Adolescent females are the demographic that makes up the largest proportion of the population participating in the recreational activities of dance and gymnastics [1,2]. These aesthetic-based sports allow the participants to maintain their physical health and offer long-term benefits such as improvement in aerobic capacity, cardio-respiratory fitness, muscle strength, and bone health [3]. Many adolescent females start at a relatively young age and take on a high volume and intensity of training. The average gymnast trains 5.36 days/week and 5.04 h/day, and some young athletes start gymnastics training as early as 4 years old, specializing in the sport at 12 years old, and performing at peak training intensity at 18 years old [4,5]. In contrast, dancers often start around as early as 3–4 years old and train 6–45 h/week, depending on age, form of dance, and technique level [6]. 

Although these sports can improve fitness, strength, and flexibility, the athletic gestures required by these sports often impose high training loads, requiring feats of great flexibility and strength, and can underlie the risk of injury [7]. Long-term, this can lead to negative health consequences, such as arthritis, decreased strength and range of motion, and limited mobility, affecting joints, soft tissues, and the growing skeleton [4,7]. A recent 10-year observational study found that female gymnasts had an injury incidence of 9.37 per 1000 athlete exposures [4]. In dance, a recent study suggested that pre-professional ballet sees 1.4–4.7 injuries per 1000 dance hours, while another study found that professional ballet sees 0.55–4.4, pre-professional contemporary dance sees 0.57–2.17, and professional contemporary dance sees 0.08–0.24 injuries per 1000 dance hours [2,8].

The primary aim of this study is to identify the risk of injury in both populations of adolescent females who participate in dance and gymnastics. Said categories include all forms of dance (recreational/professional, ballet/modern/classical, etc.) and all forms of gymnastics. Elucidating these variations could both help inform what injuries adolescent female athletes are most at risk for and provide safety recommendations to ensure the long-term health of these adolescent female athletes.

## 2. Materials and Methods

### 2.1. National Electronic Injury Surveillance System (NEISS)

A retrospective, cross-sectional analysis was performed using the NEISS database of the U.S. Consumer Product Safety Commission (CPSC) in 2020, which included the product or activity-related injuries hospital presenting to emergency departments in the U.S. No institutional review board approval was needed as the de-identified database is publicly available and published annually. Moreover, it is a nationally representative probability sample of roughly 100 U.S. emergency departments stratified by size and geographic location, from which reliable, weighted national estimates and sampling errors for queried injuries may be derived [9,10]. Variables contained in the NEISS database include the date of treatment; the case record number; the age, sex, and race/ethnicity of the patient; the injury diagnosis; the body part affected by the injury; the disposition (treated and released, admitted, etc.); the product involved (if any) in the injury; the location where the injury occurred; whether fire or motor vehicles were involved in the injury; whether the injury was work related; whether the injury was intentionally inflicted; and a short 142 character text narrative of the incident and scenario leading to the injury.

### 2.2. Patient Selection

The process of obtaining the final study population is as follows: we selected for unique dancing and gymnastics injuries in the year 2020 among females aged 18 years and younger. In this study, we queried a sample in the NEISS database from the year 2020 for injuries specifically associated with dancing and gymnastics (product codes 3278: “DANCING (ACTIVITY, APPAREL OR EQUIPMENT)” and 1272: “GYMNASTICS (ACTIVITY, APPAREL OR EQUIPMENT), N.S.”). NEISS generated an estimated number of national injuries presenting to US emergency departments by giving each injury case a statistical weight and summing up the statistical weights of each injury case [9,10].

### 2.3. Statistical Analysis

For gymnastics and dancing injuries, adjusted odds ratios (ORs) and 95% confidence intervals (CIs) were calculated using multivariable logistic regression for the outcome of predicted sport participated in (with dancing being the reference group) based on reported diagnoses. Reported diagnoses were included as a categorical variable and the outcome variable was coded as 0 or 1 with 1 corresponding to gymnastics and 0 corresponding to dancing. Specifically, the technique of multiple logistic regression was used to adjust for the potential covariates of age, weight, and body part injured. These covariates were included as continuous variables in the analysis except for body part injured which was included as a categorical variable with each level corresponding to a different body party injured. From this analysis, we were able to explore if injury significantly predicted the sport in which the adolescent women in the study participated in. *p* values < 0.05 (2-sided) were considered significant. All analyses were performed using R statistical software (version 2.14.0; www.R-project.org, accessed on 23 November 2021).

## 3. Results

### 3.1. Descriptive Statistics

Briefly, 309,370 unique patients had injuries reported in the NEISS database in 2020, reflecting a national estimate of 10,994,077 injuries. A total of 1911 cases were identified in the NEISS database during this period with dancing or gymnastics-related injuries, of which amounted to 51,708 weighted national estimates of injuries associated with both types of activities. After excluding males and patients aged 19 years or older, we found a total of 1147 unique patient cases, which amounted to 25,160 weighted national estimates of injuries associated with both types of activities. Of the 1147 unique adolescent females ages 18 or younger, 548 were among those with dancing injuries, and 604 had gymnastics-related injuries.

The annual number of adolescent female patients (18 years or younger) presenting to US emergency departments with injuries associated with gymnastics or dancing was 1152 patients in the year 2020 (Table 1). Patient demographic and injury characteristics are detailed in Table 1 with an average age of around 11 years and 14 years for those with gymnastics and dancing injuries, respectively. Average weight (in kg) and disposition of injuries were comparable across the adolescents who participated in gymnastics and dancing. In terms of distribution of race, there was a higher percentage of Black/African American patients with dancing injuries compared to gymnastics injuries (28% vs. 12%) and a higher percentage of white patients with gymnastics injuries compared to dancing injuries (56% vs. 38%). The four most common body parts injured in gymnastics included both the upper and lower extremities whereas the four most common body parts injured in dancing were all lower body extremities. Both strain/sprains and fractures were common injuries in both gymnastics and dancing, whereas contusion (7%) and laceration (6%) also appeared in gymnastics and dancing, respectively.

### 3.2. Regression Analysis

Table 2 outlines the odds ratio for the likelihood of exposure in adolescent females participating in gymnastics compared to those participating in dancing. The odds ratios reported were adjusted for age, weight, disposition, and race. Adolescent females in gymnastics had a 16% statistically significant greater probability of an upper-body injury compared to adolescent females participating in dancing [OR = 1.16 (1.01–1.38)] (Table 2). Finally, both acute and stress fractures were 3.74 times more likely in gymnastics compared to in dance, with statistical significance [OR = 3.74 (2.73, 5.18)].

## 4. Discussion

This present study was an investigation of whether differences existed in the affected body part, diagnosis, and demographics of adolescent females presenting to the emergency department with dance and gymnastics-related injuries. Dancers presented with more foot and ankle injuries, while gymnasts presented with more wrist and elbow injuries, with both findings having statistical significance. Furthermore, gymnastics injuries that presented to the emergency department were made up of a younger demographic and were overall more severe, mostly consisting of fractures. In contrast, dancing injuries mostly consisted of less severe injuries, such as strain/sprains, and were more likely to be treated in the ER and subsequently released.

### 4.1. Dance Injuries

We found that dance-related injuries presented to the emergency department most affected the knee, ankle, and foot. Fittingly, many forms of dance involve continuous lower extremity stress and require increased foot strength, ankle range of motion, and stability and control. Compared to gymnastics, there is a greater emphasis on technically demanding movements that emphasize aesthetics in certain forms of dance. In ballet, for example, en pointe, which involves maximal ankle and tarsal joint plantar flexion, increases the forces through the foot up to 12 times normal body weight [11].

It should be noted that different variations of dance have different risks, with adolescent ballet dancers facing the highest risk of injury when compared to modern dancers, older dancers, and non-dancers due to pointe work and movements that are more repetitive and rotational in nature [12,13]. Unlike gymnastics, which takes place in more regulated training areas, floor-related dance injuries—due to either slippery surfaces (dirt, liquid, props) or too much friction—are common and have been reported to make up 12.7% of accidents [14].

There is great risk for long-term injury and disability in dancers. Overall, injuries can not only lead to significant joint pain [15], but also result in early signs of hip and knee osteoarthritis in professional and retired ballet dancers [16,17]. Notably, a study looking at adult professional modern dancers found that females were not only 15 times more likely to have bone-related injuries than males but, after sustaining such injuries, females missed out on more workdays and performances [18].

### 4.2. Gymnastics Injuries

Our findings suggest that gymnastics-related injuries were associated with more fractures and were not only more likely to present to the emergency department but also more likely to be admitted into the emergency department, rather than transferred or released, than dance-related injuries. This is consistent with reports from the National Collegiate Athletic Association (NCAA) that found gymnastics to have the highest overall injury rates amongst all of women’s collegiate sports [19].

Furthermore, gymnastics-related injuries most commonly affected the lower arm, elbow, and wrist. Compressive forces resulting from gymnastic maneuvers have previously been reported to be up to 16 times an athlete’s body weight [20]. These high-impact forces can impact the wrist through torsional forces, axial compression, and distractive forces on the hyperextended joint. It can also affect the elbow through high valgus and varus stresses. The ligaments surrounding joints often undergo plastic deformation following continuous microtrauma, resulting in increased laxity and instability [21,22].

### 4.3. Injury Prevention

Injuries stem from three sources: growth factors during adolescence, intrinsic factors like previous injury or menstrual irregularities, and extrinsic factors such as training environment or technique. Previous research suggests that adolescent athletes are more prone to injuries than adult athletes, as they face risk factors such as a continually developing skeleton and underdeveloped coordination, spatial perception, and limbs that are constantly getting longer, all of which interfere with their neuromotor control and stability [23,24,25,26]. As performance-based sports, dance and gymnastics also emphasize an additional aesthetic component in which performances are partially evaluated by their degree of artistry or style [27]. This emphasis on body image favors adolescent females that have low body fat mass. Often, these females maintain extreme calorie deficits that can lead to psychiatric complications and eating disorders that can exacerbate the physical demands of each sport and limit normal pubertal development in adolescents [28].

Many studies have found that, overall, female athletes experience higher rates of stress fractures than male athletes [29,30]. In gymnastics, for instance, 24.4% of women required surgery after injury compared to 9.2% of men [31]. The phenomenon of relative energy deficiency in sport (RED-S) has been said to occur in both female and male athletes—in sports with similar emphasis on aesthetics and/or weight, with the majority of RED-S cases in female athletes [32,33]. Furthermore, adolescent females have less muscle mass, thought to be protective against stress fractures, as well as shorter height and greater joint laxity compared to adolescent males [34,35].

Both gymnastics and dance have a high rate of early single-sport specialization, which has previously been associated with overuse injuries [36,37]. Notably, adolescents between the ages of 6 and 15 are more likely to present to the ED with gymnastics-related injuries than adolescents over 15; this is thought to stem from the increased pressure to attempt more challenging new skills from a young age to maintain competitiveness [38]. Most dance-related injuries are overuse injuries, while gymnastics-related injuries more commonly arise from trauma [39]. Unlike other sports, gymnastics requires athletes to perform maneuvers that place unique biomechanical stresses on parts of the body, requiring flexibility in highly controlled twisting, rotating, and swinging in the air followed by static holds, rebounds, and hard landings [40,41].

For young pre-professional athletes, injury can be debilitating, not only impacting their future career prospects within the sport but also their mobility and long-term quality of life. There are many causes of injuries, which include improper training, faulty technique, environmental hazard, anatomical structural deformities, and biomechanical imbalance [42]. Injuries stemming from the first three can be targeted and avoided. Often, these injuries result from improper training and faulty technique: the absence of warm-up routine, repetitive jumping, poor body weight alignment, and disregarded overuse/fatigue. As early injury detection and treatment is critical to decreasing injury severity, it is imperative that routine injury surveillance is implemented, with special attention to the growth plate in adolescents [43]. Ultimately, return to sport is following complete recovery of both strength and range of motion.

Studies have shown that increased psychological and physiological stress can affect the number and severity of injuries as well as slow down the time of recovery [44,45]. Fatigue is a prominent risk factor for injuries, as it often leads to altered motor control strategies [46]. Additionally, the buildup of microtrauma that can occur over a high volume of training can result in severe injury [41,47]. As indoor sports, athletes typically train year-round without a dedicated rest period. Furthermore, studies have found that adolescent athletes with more intensive practice schedules (greater than 10 h a week) tend to sleep less and are thus deprived of valuable recovery time and may be more prone to injury due to cumulative fatigue. We recommend regularly scheduled breaks between practices to rest and recover to prevent fatigue and minor injuries from accumulating for maximal physiological longevity [48].

In order to prevent injuries, we recommend greater safety protocols involving adequate warming-up, physical conditioning, proper equipment, and technical regulations for each sport. With proper form and frequency, training can improve strength, endurance, power, agility, plyometrics, balance, and joint stability [43]. For dance specifically, we recommend more regulation with dance studio facilities and footwear. Additionally, athletes face a nearly 2-fold increase in their risk of injury when competing compared to when practicing [3]. Furthermore, the difficulty with implementing safety recommendations in gymnastics reflects the presence of maneuvers with higher technical difficulty that are scored higher in competition. To dissuade young athletes from trying maneuvers they are not yet ready to attempt, Hart et al. have suggested that greater emphasis could be placed on maneuver execution and greater deductions for falls during competition [49]. It must be noted that safety requirements for equipment and greater breaktime in between routines should be maintained, even at more formal levels of competition [18].

### 4.4. Strengths and Limitations

A strength of this study comes from the publicly accessible data that NEISS draws from. Basing consumer product-related injuries on nationally representative probability samples of hospitals increases the generalizability of findings from this study. There are limitations to using the NEISS dataset, which only captures injuries that were presented to a representative sample of 100 emergency departments across the US. The data presented are limited in nature, as only a brief case narrative is recorded for each encounter. Further, only one hospital encounter is recorded per individual, so calculating measures related to incidence would not be possible. NEISS does not distinguish between the different forms of dance and gymnastics, within which there may be a large range of injury profiles. Additionally, the NEISS database does not include the length of experience a particular athlete has in the sport, only that they are participating in the sport. Thus, it is difficult to make conclusions about potential associations between injury patterns and expertise within each sport.

There is also an inherent sampling bias, as patients could have gone to urgent care, their primary care provider, physical therapist, or to an orthopedic surgeon. Furthermore, given the anonymity of the patient data set, there is no way to correct for patients who could have been evaluated more than once. Another weakness to consider is that covariate data collected from participants could be subject to measurement error with respect to the potential misreporting of variables such as weight and height. Additionally, conclusions about the overall context (practice, during competition, recreational, etc.) of the injury cannot be made using NEISS given the variable information, if included, in the free text section of each injury report. Limiting the number of participants in the study to those that had thorough information on the context of each injury would reduce the power. Future studies could examine differences in age-related injury patterns within each of the two sports to further increase prevention strategies that would be specifically tailored to specific age groups within each sport.

Residual confounding by covariates not evaluated was possible. This study undertook an observational design without the ability to minimize the confounding effect of unmeasured factors as well as randomized controlled trials can. Health-seeking behaviors, such as length of time after onset of injury, were not accounted for in this study. Sensitivity analyses such as excluding participants who have had previous injuries to the same or different body part were not explored. Future studies could examine the extent of residual confounding by including more covariates that account for the past medical history of each patient.

Finally, the role of chance cannot be dismissed from consideration. Multiple comparisons between categories of body parts injured and types of injuries and sports participated in were made with no adjustment of the significance level. Since statistical power depends on the number of occurrences, this study population may have been underpowered to detect a statistical association between the type of injury and sport participated in.

Another limitation is that injuries coded as dance or gymnastics-related injuries do not just involve elite-level athletes, but also recreational athletes and adolescent females who do not identify as athletes that are participating in these sports. This makes it harder to generalize conclusions around the findings of this paper to just adolescent female athletes.

## 5. Conclusions

To our knowledge, this is the first large-scale study to investigate and compare the musculoskeletal injury patterns of gymnastics and dance-related injuries presented to the emergency department. More adolescent females presented to the emergency department with gymnastics-related injuries than dance-related injuries. Dance is associated with more sprains, particularly in the knee, while gymnastics typically results in a greater likelihood of fractures, particularly in the joints of the lower arm, which does not include the wrist or hand.

We recommend engaging in therapeutic exercise including adequate warming-up and physical conditioning for injury prevention. For both gymnastics and dancing, there should be proper safety equipment in place both during practices and performances. Technical regulations as well as routine injury surveillance should be implemented to ensure early injury detection and treatment. Additionally, preventative strengthening and conditioning programs should be explored for more injury-prone areas of the body for both dancers and gymnastics. When injuries do arise, formalized rehabilitation programs should be studied and formalized to streamline the recovery process.

The findings presented are not intended to deter adolescent females from participating in gymnastics and dance. Rather, this information should be taken into consideration by parents when deciding on what sport to engage in and how to do so in a safe and effective manner that optimizes long-term physical health and proficiency in said sport. The unique risk factors for each sport can be used to tailor injury prevention strategies to make these sports safer. Further study to build on these findings and identify specific drivers of injury within each sport is needed.

## Figures and Tables

**Table 1 ijerph-20-00471-t001:** Descriptive characteristics of adolescent females with gymnastics and dancing-associated injuries reported to the U. S. NEISS in 2020.

Variables	Gymnastics (N = 604)	Dancing (N = 548)	Total (N = 1152)
Age in years, mean (SD)	11 (12)	14 (21)	13 (18)
Race, N (%) [95% CI] ^1^			
Not stated	162 (27%) [21.3–33.2]	166 (30%) [28.2–22.4]	328 (28%) [27.5–30.0]
White	340 (56%) [54.2–59.1]	209 (38%) [36.2–39.4]	549 (48%) [46.5–52.4]
Black/African American	74 (12%) [9.5–12.1]	151 (28%) [24.6–29.5]	225 (20%) [15.5–24.6]
Other	21 (3%) [1.4–3.1]	12 (2%) [1.6–3.2]	33 (3%) [0.5–5.0]
Asian	7 (1%) [0.4–2.1]	6 (1%) [0.5–3.0]	13 (1%) [0.5–2.5]
Weight in kg, mean (SD)	22 (27)	22 (26)	22 (27)
Disposition, N (%) [95% CI] ^1^			
Treated and released, N (%)	566 (94%) [90.8–96.2]	536 (98%) [96.2–98.4]	1101 (96%) [93.8–99.1]
Treated and transferred, N (%)	4 (0.7%) [0.3–1.4]	1 (0.2%) [0.2–1.4]	5 (0.4%) [0.2–1.0]
Treated and admitted, N (%)	31 (5%) [2.1–6.7]	5 (0.9%) [0.2–3.2]	36 (3%) [1.1–5.4]
Held for observation, N (%)	1 (0.17%) [0.12–0.4]	1 (0.2%) [0.1–0.4]	2 (0.2%) [0.1–0.4]
Left without being seen, N (%)	2 (0.33%) [0.11–0.78]	5 (0.9%) [0.10–4.2]	7 (0.6%) [0.1–4.2]
Four most common body parts injured, N (%) [95% CI] ^1^			
	Lower arm, 74 (12%) [8.2–14.7]	Knee, 125 (23%) [18.2–27.8%]	Knee, 160 (14%) [10.2–17.8]
	Elbow, 70 (12%) [7.5–23.6]	Ankle, 73 (13%) [8.4–14.7]	Ankle, 132 (11%) [8.9–15.6]
	Wrist, 52 (9%) [4.6–10.5]	Foot, 52 (9%) [5.6–13.1]	Lower arm, 94 (8%) [6.7–14.5]
	Ankle, 50 (8%) [5.7–12.1]	Lower Trunk, 37 (7%) [4.4–10.1]	Wrist, 72 (6%) [3.5–10.8]
Four most common injury diagnoses, N (%) [95% CI] ^1^			
	Fracture, 224 (37%) [32.3–40.2]	Strain/Sprain, 180 (33%) [31.2–37.7]	Strain/sprain, 312 (27%) [24.2–30.2]
	Strain/Sprain, 132 (22%) [17.1–25]	Other/Not Stated, 155 (28%) [24.1–29.8]	Fracture, 295 (25%) [20.1–27.6]
	Other/Not Stated, 120 (20%) [15.2–22.3]	Fracture, 71 (13%) [10.1–16.9]	Other/Not Stated, 275 (24%) [20.2–28.3]
	Contusion, 45 (7%) [4.6–10.1]	Laceration, 32 (6%) [3.4–9.8]	Contusions, 76 (6%) [2.3–12.4]

^1^ 95% CI for percentages provided in the table.

**Table 2 ijerph-20-00471-t002:** Odds ratios for injuries in in gymnastics compared to dancing *.

Variables	Unadjusted OR (95% CI) [*p* Value]	Adjusted OR (95% CI)
Age	0.90 (0.70–1.10) [0.20]	0.93 (0.72–1.13) [0.22]
Weight	0.99 (0.95–1.05) [0.08]	0.98 (0.95–1.05) [0.08]
Body part injury (upper extremity vs. lower extremity)	1.20 (1.02–1.40) [0.04]	1.16 (1.01–1.38) [0.03]
Strain/sprain	0.88 (0.66, 1.16) [0.30]	0.87 (0.65, 1.15) [0.28]
Fracture	3.78 (2.77, 5.21) [0.01]	3.74 (2.73, 5.18) [0.01]

* Reference group is dancing.

## Data Availability

Publicly available datasets were analyzed in this study. This data can be found here: at https://www.cpsc.gov/ (accessed on 19 November 2021).

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
