# Peer review of "Comparing Musculoskeletal Injuries across Dance and Gymnastics in Adolescent Females Presenting to Emergency Departments"

_ijerph, 2022, doi:10.3390/ijerph20010471_

Round 1
Reviewer 1 Report
Dear authors, the manuscript has some insight. I have concerns about the drafting of the paper and possibly the clinical impact of its suitability for publication. However I leave to the editor, any decision on the matter
Was there a certain practiced experience in the abstract as inclusion criteria?
31 reference missing
44 reference missing
I might suggest something like that “therefore, the athletic gestures required by these sports very often impose high training loads and can underlie the risk of injury.”
Ref: Marotta, N., Demeco, A., Moggio, L., Isabello, L., & Iona, T. (2020). Correlation between dynamic knee valgus and quadriceps activation time in female athletes. Journal of Physical Education and Sport, 20(5), 2508-2512 http://doi.org/10.7752/jpes.2020.05342
44-46 however the exposure for athletes and dance hour are different conceptions of incidence
59 the methods are completely unstructured.. there is the sample selection flowchart which is strictly a result.. move the entire paragraph into the results
Design, but above all participants? Each inclusion and exclusion criterion must be defined before proceeding with the selection
Outcome, but above all the statistical analysis section is missing. Have you evaluated the frequency and prevalence. What statistical tests have been conducted?
112 is not significant with this confidence interval (0.65, 1.15).. remove or not highlight the result
Author Response
Please see attached Word document

Reviewer 2 Report
This is a cross-sectional study using data collected from patients presenting to U.S. emergency departments participating in the publicly available, identified U.S. Consumer Product Safety Com-16 mission's National Electronic Injury Surveillance System (NEISS) during the year 2020
Reviewer's remarks:
In the material and methods section or the introduction, it should be explained why it was decided to analyze the year 2020. Perhaps an analysis of individual years of the last decade was carried out and this year turned out to be special. This should be explained.
In the conclusion section the authors recommend engaging in therapeutic exercise, including adequate warm-up and physical preparation to prevent injury.
On what basis do they conclude that there is no such involvement and treat it as a cause of injury?
Data in this area have not been obtained and analyzed.
Author Response
Please see attached Word document

Reviewer 3 Report
The topic is interesting and provide some useful information regarding the comparison of injuries across adolescent female dancers and gymnasts. Nevertheless, I have a fundamental question that could affect the appropriateness of study design and the results of the study:
Are there any differences in injury prevalence at different ages. It is important to investigate age-based patterns in youth gymnastics–related injuries. E.g. a great majority of injury are observed in children and young teenagers, demonstrating that a significant injury burden is present among young gymnasts.
Do injuries occur during practice or in competition?
Does NEISS contain Exposure data? I have not seen any information regarding injury rates (annual number of injuries per 100,000 exposures -athlete-days- calculated with the use of provided statistical weights.
Abstract.
The term ‘adolescent females’ is too general, please add the age of the participants.
Introduction and Conclusion
I suggest citing more recent literature.
See, for example:
Tisano et al. Epidemiology of Pediatric Gymnastics Injuries Reported in US Emergency Departments
Sex- and Age-Based Injury Patterns. The Orthopaedic Journal of Sports Medicine, 10(6), 2022
Albright et al. Characterization of Musculoskeletal Injuries in Gymnastics Participants From 2013 to 2020. Sports Health. 2022 Jun 7;19417381221099005. doi: 10.1177/19417381221099005.
Critchley et al. Injury epidemiology in pre-professional ballet dancers: A 5-year prospective cohort study. Physical Therapy in Sport 58 (2022) 93-99
Caroline McBride & Shaw Bronner. Injury characteristics in professional modern dancers: A 15-year analysis of work-related injury rates and patterns. JOURNAL OF SPORTS SCIENCES 2022, VOL. 40, NO. 7, 821–837
Hart et al. The Young Injured Gymnast: A Literature Review and Discussion. Current Sports Medicine Reports: November 2018 - Volume 17 - Issue 11 - p 366-375.
Introduction, line 45. There is a typo error. Please correct (2) in [2].
Author Response
Please see attached Word document

Round 2
Reviewer 1 Report
Dear Authors, the manuscript can provide an intriguing epidemiological picture of the this interesting topic, however I suggest greater methodological rigour..
85 Population. I renew the recommendation to move the results of the selection in the proper section, in the methods it is appropriate to describe the whole methodological path, from research, to the stratification of the strategy but without results.
There are no cut-offs at least for age, there are elderly subjects who practice dance, but I don't think this is the case. In reality, if this were the case, I recommend reporting it in the methods (in fact, on line 90, you sketch some exclusion criteria, but they should be better defined and stated at the beginning of the paragraph)
Was there a validation, among other things? In the sense that two authors conducted the research in parallel and if there were disagreements, a third author (?) was consulted?
Thank you for adding the section, but the term regression is vague at the beginning of the paragraph. You should report how you are going to describe the data.. as continuous and frequency etc.
109 This is a method, you're describing eligibility. This statement should be moved to the methods, the previous whole selection flowchart should be placed here, among other things with a figure for greater clarity (just a suggestion, not a reccomandation)
Table 2. I suggest to check, it seems strange to me that some confidence intervals that do not include 1, are not significant
183 Please reduce this paragraph as it is not within your scope, they are suggestions in the light of the literature. But you need to discuss the results of your study. I suggest the same in the conclusions paragraph, it is not inappropriate to give suggestions on the subject, but they cannot be greater than the conclusions of your study itself.
Author Response
See word document attached.

Reviewer 3 Report
The Authors have taken into account all the observations I have made, so in this form the manuscript is acceptable for publication.
Author Response
Thank you for taking the time to review our paper. I hope that you found our edits satisfactory.
Round 3
Reviewer 1 Report
Dear Authors, I defer to the Editor's decision. I would suggest reducing the claims about injury prevention.. they don't fit into the study topics.
Author Response
See word document
